# Carcinogenesis and Metastasis: Focus on TRPV1-Positive Neurons and Immune Cells

**DOI:** 10.3390/biom13060983

**Published:** 2023-06-13

**Authors:** Nuray Erin, Arpad Szallasi

**Affiliations:** 1Department of Medical Pharmacology, School of Medicine, Akdeniz University, Antalya 07070, Turkey; 2Immuno-Pharmacology and Immuno-Oncology Unit, School of Medicine, Akdeniz University, Antalya 07070, Turkey; 3Department of Pathology and Experimental Cancer Research, Semmelweis University, H-1085 Budapest, Hungary

**Keywords:** TRPV1, thermoTRP, carcinogenesis, neuroimmune regulation, capsaicin, resiniferatoxin

## Abstract

Both sensory neurons and immune cells, albeit at markedly different levels, express the vanilloid (capsaicin) receptor, Transient Receptor Potential, Vanilloid-1 (TRPV1). Activation of TRPV1 channels in sensory afferent nerve fibers induces local effector functions by releasing neuropeptides (most notably, substance P) which, in turn, trigger neurogenic inflammation. There is good evidence that chronic activation or inactivation of this inflammatory pathway can modify tumor growth and metastasis. TRPV1 expression was also demonstrated in a variety of mammalian immune cells, including lymphocytes, dendritic cells, macrophages and neutrophils. Therefore, the effects of TRPV1 agonists and antagonists may vary depending on the prominent cell type(s) activated and/or inhibited. Therefore, a comprehensive understanding of TRPV1 activity on immune cells and nerve endings in distinct locations is necessary to predict the outcome of therapies targeting TRPV1 channels. Here, we review the neuro-immune modulation of cancer growth and metastasis, with focus on the consequences of TRPV1 activation in nerve fibers and immune cells. Lastly, the potential use of TRPV1 modulators in cancer therapy is discussed.

## 1. Introduction: TRPV1, Capsaicin, and Neurogenic Inflammation

Natural products provide a window of opportunity to discover novel molecular pathways. A prominent example is capsaicin, the pungent ingredient in hot chili peppers. Hot pepper is indeed “hot” because capsaicin and noxious heat activate the very same receptor on sensory neurons [1], now known as Transient Receptor Potential, Vanilloid-1 (TRPV1) [2]. TRPV1 is the founding member of the family of temperature-sensitive TRP channels, collectively referred to as “thermoTRPs” [3]. Combined, thermoTRPs cover a broad range of temperatures, from noxious hot to noxious cold [4]. The central role of TRPV1 in noxious heat sensation was highlighted by the burn injuries that participants of clinical trials with TRPV1 antagonists suffered as adverse effect [5]. The 2021 Nobel Prize in Medicine and Physiology was awarded to David Julius (shared with Ardem Patapoutian), in part for discovering the molecular mechanisms of heat sensation that started with the molecular cloning of TRPV1 in 1997 [1].

Capsaicin represents an area of sensory pharmacology intimately familiar to those of us who enjoy eating hot, spicy food. Capsaicin causes a burning sensation on the tongue that dissipates (“desensitizes”) upon repeated challenge [6,7]. Animals desensitized to capsaicin are also unresponsive to various unrelated chemical and physical stimuli [8]. The therapeutic potential of capsaicin desensitization was recognized early. High-dose capsaicin patches [9] and site-specific injections [10] are already in clinical practice to relieve chronic neuropathic pain. The molecular mechanisms of capsaicin desensitization are, however, largely unknown.

TRPV1 is predominantly expressed on sensory neurons. Indeed, a whole subdivision of primary sensory neurons is named after their unique sensitivity to capsaicin [11,12,13].

Capsaicin-sensitive (TRPV1-expressing) sensory neurons are bipolar cells with cell bodies in sensory (dorsal root and trigeminal) ganglia. A subset of vagal neurons with somata in nodose ganglia also express TRPV1 [14]. Generally speaking, TRPV1-expressing neurons detect harmful information in the periphery and convey this information into the central nervous system (CNS). Importantly, these neurons not only detect noxious stimuli as an afferent function, but they also produce and release pro-inflammatory neuropeptides from their peripheral endings as a local efferent function [15]. Notable neuropeptides include substance P (SP) and calcitonin gene-related peptide (CGRP). These neuropeptides trigger the biochemical cascade known as neurogenic inflammation [16,17]. The role of neurogenic inflammation in various disease states (from migraine to diabetes) is well beyond the scope of this review (interested readers are referred to [17]. Here, it suffices to mention that neurogenic inflammation affects carcinogenesis [18], tumor growth [19], and metastasis formation [20].

Albeit at much lower densities than on capsaicin-sensitive sensory neurons, TRPV1 is expressed in various non-neuronal cells, including immune cells [21]. This implies that TRPV1 agonists (or antagonists) may produce complex and unpredictable effects by simultaneously acting on neuronal and non-neuronal targets. Moreover, neuropeptides released from capsaicin-sensitive nerve endings can also influence the activity of both cancer cells and immune cells [22]. For instance, SP may enhance anti-tumoral activity by triggering dendritic cell migration [23] and augmenting antigen-induced lymphocyte proliferation [24]. Conversely, SP can stimulate the proliferation of cancers that express the SP receptor, NK-1 [25]. Take glioblastoma cells as an example: they express both TRPV1 [26] and NK-1 receptors [27]. On the one hand, capsaicin may induce apoptosis in this tumor by activating TRPV1 [26]. On the other hand, capsaicin may accelerate tumor growth by liberating SP from sensory nerves that, in turn, stimulates trophic NK-1 receptors in cancer cells [26]. Indeed, NK-1 receptor antagonists were proposed to represent a new therapeutic approach in glioblastoma patients [28].

## 2. Neurogenic Inflammation and Tumor Promotion in the Skin

165 years ago, Robert Virchow in his “Reiztheorie” (inflammation theory) postulated a causative connection between inflammation and cancer based on his observation that white blood cells are found in increased numbers in the vicinity of tumors [29]. In support of this theory, the first experimental proof was provided by Yamagiwa and Ichikawa by painting coal tar on the ear of rabbits [30]. The cancer causing agent in coal tar was later identified as 7,12-dimethylbenz[a]anthracene (DMBA). DMBA as a single carcinogen can induce the formation of epithelial tumors when applied to the back skin of mice [31,32]. The tumor-causing dose of DMBA, however, can be greatly reduced if croton oil were applied subsequent to the application of a dose of DMBA too low to induce tumors by itself [32,33]. In this classic two-stage carcinogenesis experiment, DMBA is thought to initiate the tumor development by mutagenic activity, whereas croton oil is believed to promote the formation of tumors by maintaining chronic inflammation.

Based on this seminal observation, the mouse ear erythema assay was routinely used to detect tumor-promoting compounds in natural product extracts. In fact, a close relationship was postulated between the inflammatory and tumor promoting activities of natural products [34]. The prototypical tumor promoter, 12-O-tetradecanoylphorbol-13-acetate (TPA), is indeed a very potent inflammatory agent in the mouse skin [35]. TPA is the main ingredient in croton oil, the seed oil of the Euphorbiaceae *Croton tiglium*. Using the mouse ear erythema assay, another extremely irritant compound was discovered in the soap of *Euphorbia resinifera* Berg [36], named after its source: resiniferatoxin (RTX). Unlike TPA, RTX does not promote the formation of tumors [37]. Although TPA and RTX use completely different molecular pathways to cause inflammation [38], protein kinase-C (PKC) [38,39] and TRPV1 [40], respectively, both pathways participate in neurogenic inflammation [41].

In CD1 mice (a mouse strain particularly sensitive to tumor promotion by TPA), desensitization by RTX of the neurogenic inflammatory response completely prevented the early (3 h) erythema and edema response to TPA, and partially blocked the edema at later times (up to 24 h) [41]. However, RTX desensitization had no measurable effect on TPA-induced epidermal hyperplasia and/or tumor formation [41]. These findings imply that tumor promoters indeed do evoke neurogenic inflammation, but it is the non-neurogenic component of their inflammatory reaction that promote the formation of skin tumors. A similar conclusion was reached in mouse strains with different susceptibilities to phorbol ester tumor promotion [18].

Human keratinocytes express functional TRPV1 [42]. Activation by capsaicin of keratinocyte TRPV1 triggers the release of the pro-inflammatory mediators, interleukin-8 and prostaglandin-E2 [43]. This observation raises the question whether capsaicin can promote skin carcinogenesis via non-neurogenic inflammation. In the back skin of ICR mice, topical capsaicin (a single application of 10 µmol) followed by TPA twice a week resulted in no increase in the number of skin tumors compared to solvent controls [44]. Following initiation by DMBA, RTX did not promote the formation of skin tumors either [37]. Last, no difference was noted in skin tumor development between wild-type or solvent control animals and those whose TRPV1 had been ablated by genetic manipulation (TRPV1 KO mice) [45] or neonatal RTX administration (A. Szallasi and P.M. Blumberg, unpublished observations).

Combined, these observations imply that (1) RTX by itself is not a complete carcinogen and (2) the elimination by RTX of the neurogenic inflammatory response does not modify the response in the two-stage carcinogenesis model.

## 3. Neuro-Immune Regulation of Cancer

Interaction between the immune system and the CNS is crucial for maintaining homeostasis. Abnormal activation and/or disturbance of this neuro-immune interaction may have serious pathological consequences [46]. Chronic, stress-induced alterations in immune responses may have detrimental consequences for cancer patients [47]. For example, chronic or disproportionate stress may suppress protective immunity, induce or exacerbate chronic inflammation, or enhance immuno-suppressive mechanisms [47,48]. Early psychosocial intervention may significantly reduce cancer mortality [49].

The nervous system plays a crucial role in carcinogenesis, particularly through altering anti-cancer immunity (Figure 1) [50]. Multiple components of neuro-immune interactions exist, including the adrenergic sympathetic nerves, the hypothalamic–pituitary–adrenal axis, and the parasympathetic nerves [51,52,53]. The sympathetic nervous system was shown to promote the formation of solid tumors [54]. Adrenergic sympathetic nerves directly innervate all primary and secondary immune organs [55], and the excessive activation of these nerves can suppress anti-tumoral immunity. By contrast, the activation of the parasympathetic system was shown to exert anti-tumoral and anti-metastatic effects via enhancing anti-tumoral immunity [49,56]. Specifically, activation of parasympathetic nerves can downregulate the expression of programmed cell death protein-1 (PD-1) on CD4+ tumor-infiltrating T-lymphocytes, and improve the expression of interferon-γ, thus enhancing anti-tumor immunity [56].

The cholinergic parasympathetic system Innervates the majority of visceral organs directly via the vagal nerve or through the celiac ganglia (Figure 1). The parasympathetic efferent fibers terminate in the sympathetic celiac ganglia [57,58]. Approximately 80% of nerve fibers in the vagal nerve are afferent (sensory) fibers that carry sensory information from the visceral organs to the brain [59]. In turn, efferent fibers of the vagal nerve transmit central neuronal response into the visceral tissues as well as innate and adaptive immune cells [60,61]. The vagus nerve mediates neuro-immune interactions in cancer [62], in addition to its fundamental physiological regulatory functions in controlling circulation, temperature, digestion, respiration, and the immune response [63,64,65,66].

## 4. TRPV1-Positive Vagal Afferent Neurons and Carcinogenesis

Vagal visceral afferent nerve endings express TRPV1 [14], a non-selective cation channel allosterically modulated and/or activated by a range of thermal, mechanical and chemical stimuli [67,68]. TRPV1 activation may mediate the immune regulatory role of the vagus nerve. Peripheral activation of vagal afferent fibers carries the information to the nucleus tractus solitarii in the brainstem. In return, activated efferent vagus nerve fibers inhibit the production of peripheral pro-inflammatory cytokines through a nicotinic anti-inflammatory pathway [69,70]. In accord, in a concanavalin-A model of hepatitis, the afferent vagal nerve activity is required for the anti-inflammatory response in the liver of treated animals [71].

In experimental breast cancer, afferent vagus nerve fibers mediate the anti-metastatic effect of vagal activity. Chemical (by systemic capsaicin) or surgical vagotomy results in enhanced metastasis formation [20,72,73,74]. Conversely, activation of the vagus nerve by semapimod (a non-steroidal inhibitor of inflammatory cytokine production) decreases breast cancer metastasis [75]. Moreover, vagal stimulation reduces the levels of pro-inflammatory cytokines (e.g., interleukin-1β and tumor necrosis factor-α), and enhances anti-tumor immunity by activating cytotoxic CD8+ T cells and NK cells [76]. Vagal stimulation also inhibits the accumulation and activity of myeloid-derived suppressor cells in breast cancer [77]. Upon activation, afferent vagal fibers release SP, which, in turn, enhances anti-tumoral immunity by increasing the cytotoxic effect of lymphokine-activated NK cells [78]. SP also increases interleukin-12 secretion and promotes the maturation of dendritic cells [79].

In metastatic breast carcinoma, continuous SP treatment by implantable pumps markedly enhances the therapeutic efficacy of radiotherapy by increasing the anti-tumoral immune response [80]. Hence, stimulation of vagal afferent nerve fibers via TRPV1 activation may inhibit chronic inflammatory responses and enhance anti-tumoral immunity, and thus establish an anti-tumoral environment. In keeping with this hypothesis, olvanil, a non-pungent vanilloid agonist that activates TRPV1 more slowly than capsaicin does [81,82], was shown to suppress metastasis of breast carcinoma without altering the rate of primary tumor growth [83]. Olvanil also decreases the systemic inflammatory response and enhances anti-tumoral immunity at a dose that primarily activates the nerve fibers [83].

## 5. Systemic Activation of Neuronal TRPV1 in Cancer

TRPV1 is expressed in both sensory nerve fibers and non-neuronal tissues, though expression levels differ markedly between neuronal and non-neuronal cells. Nanomolar concentrations of capsaicin (and related TRPV1 agonists) are sufficient to activate nerve fibers [84]. By contrast, over 30 μM capsaicin concentrations are required for TRPV1 activation in isolated immune cells [85]. Hence, selectivity of TRPV1 agonists at lower concentration is much higher for sensory nerve fibers. For example, the affinity of capsaicin for the human neuronal TRPV1 channel ranges from 7 to 30 nM as determined through electrophysiological studies [86,87]. By contrast, capsaicin stimulates human monocytes via TRPV1 at concentrations exceeding 100 µM [85]. Therefore, at lower doses TRPV1 agonists are likely to affect mainly the peripheral nervous system.

TRPV1 is thought to play an important role in the pathology of autoimmune neuritis [88]. In rats, low doses of orally administered capsaicin ameliorated the symptoms of autoimmune neuritis [89]: given at a dose of 50 μg/day, corresponding to consuming one or two chili peppers per day in humans, dietary capsaicin was already effective [89]. The beneficial effect of capsaicin in autoimmune neuritis was attributed to the release of sensory neuropeptides, such as CGRP and SP, which in turn can inhibit macrophages. A similar protective effect of low capsaicin doses was reported in an autoimmune diabetes model [90]. In keeping with the animal experiments, in a large-scale prospective cohort study, daily consumption of spicy (chili-based) food was inversely associated with all-cause mortality, and specific-cause mortality including cancer [91].

The TRPV1-mediated anti-tumoral effect of capsaicin can be attributed to its anti-inflammatory actions. This assumption is supported by *Trpv1* gene knockout studies. In a mouse model of allergic contact dermatitis, exacerbated inflammatory reaction was observed in the TRPV1 null animals [92]. In this model, TRPV1 deficiency is associated with the upregulation of pro-inflammatory cytokines, increased infiltration of macrophages, and simultaneous inflammation [92]. Similarly, more severe inflammatory response occurs in TRPV1-null mice challenged with LPS to induce shock and organ damage (renal and hepatic) [93]. Moreover, compared to wild-type littermates, TRPV1-null mice exhibit more severe LPS-induced sepsis [94], and more often develop colitis-associated cancer [95]. In accord, small-molecule TRPV1 agonists were shown to reduce inflammation in sepsis models [96,97,98].

## 6. Role of the Central Nervous System (CNS) in the Anti-Inflammatory Effects of TRPV1 Agonists

There is growing evidence that TRPV1 agonists may target CNS nuclei, such as the hypothalamus, to modulate inflammatory responses. TRPV1 channels are expressed in the brain, particularly in areas that are responsible for controlling the autonomic output to the visceral organs (Figure 1). For example, in rat brain, TRPV1 channel expression was demonstrated in the paraventricular nucleus of the hypothalamus and the dorsal motor nucleus of the vagus [99]. Using a viral tracer, TRPV1 receptors were also found in liver-related pre-autonomic neurons [100]. In addition to the hypothalamus, TRPV1 is expressed in the hippocampus and midbrain [101,102]. This restricted TRPV1 expression in the brain is conserved across species (rat, monkey, and human) [101,102,103].

TRPV1 can modulate synaptic transmission via both pre- and postsynaptic mechanisms. Depending on the location, TRPV1 may participate in various (sometimes opposing) functions [104]. Expressed pre-synaptically on afferents to the locus ceruleus, TRPV1 potentiates the release of glutamate and adrenaline/noradrenaline [105]. By contrast, TRPV1 suppresses the excitatory transmission in the dentate gyrus via a postsynaptic mechanism [106,107,108]. Importantly, TRPV1 is expressed in microglia and astrocytes [109,110]. In fact, microglial TRPV1 is thought to mediate some central effects of TRPV1 agonists, such as the enhancement of neuronal glutamatergic transmission [111]. In the substantia nigra, TRPV1 activation in astrocytes produces ciliary neurotrophic factor: this rescues nigral dopaminergic neurons in experimental Parkinson disease [112].

“Endovanilloids” (endogenous TRPV1 agonists generated in the CNS [113]) may exert potent anti-inflammatory activity. In models of endotoxemia and polymicrobial sepsis, the potent endovanilloids, N-arachidonoyl dopamine (NADA) and N-oleoyl dopamine, reduced inflammatory responses [114,115,116]. These endovanilloid effects were mediated by TRPV1 expressed in the CNS and not in the peripheral nervous system [114].

Capsaicin is highly lipophilic: it can pass the blood–brain barrier easily and has a longer half-life in the brain compared to peripheral tissues [117]. Therefore, the anti-tumoral effects of TRPV1 agonists might partly be due to activation of TRPV1 in CNS nuclei, such as the hypothalamus, that regulate vagal activity (Figure 1).

Although the local anti-tumoral effects of TRPV1 agonists are well-documented, the systemic effects of TRPV1 agonists on carcinogenesis and metastasis vary depending on the model. For instance, TRPV1 activation may actually accelerate cancer growth [20,118]. Below, the possible causes of these discrepancies are discussed, focusing on non-neuronal TRPV1 activation, as well as the neuronal components of the tumor microenvironment.

## 7. Innervation of the Cancer

Once solid tumors were thought to lack innervation [119]. This is now known to not be true. In fact, tumor cells can produce their own innervation, a process termed by Palm and Entschladen as “neo-neurogenesis” [120]. Palm and Entschladen also introduced the concept of “neuro-neoplastic synapsis”, defining the structures that enable interaction between peripheral neurons and tumor cells [119,120].

Further studies documented the presence of a heterogeneous group of nerve fibers within tumor tissue that is often associated with the aggressiveness [121,122]. In breast and prostate cancers, increased nerve densities are associated with more aggressive tumor phenotype and poor patient survival [121,123,124]. Cancer cells can produce growth factors, such as nerve growth factor (NGF), and stimulate the growth of new axons into the tumor [125,126]. The type of nerve fibers might be critical in determining the aggressiveness of the cancer. For example, in a chemically induced breast tumor model, sympathetic nerve activation may accelerate cancer growth and progression, while selective parasympathetic nerve fiber activation may conversely inhibit tumor growth [56]. A strong connection between the autonomic nervous system and a tumor was also demonstrated in a mouse model of breast cancer [124]. Genetic depletion of intra-tumoral sympathetic nerves inhibits breast cancer growth [56,127]. To promote growth and metastasis, cancer cells can manipulate the nervous systems by stimulating the growth of new sympathetic nerve fibers into the tumor tissue, and by trans-differentiating the phenotype of sensory neurons into an adrenergic type [128,129]. Indeed, in patients with recurrent breast cancer, the sympathetic nerve density was higher and the parasympathetic nerve density was lower compared to patients without recurrent breast cancer [56].

Among cancer tissue and the CNS, indirect connections were described in several tumor models, including fibrosarcoma [130] and melanoma [131]. Recent studies also show a direct link between cancer and the CNS. For example, using electrical nerve stimulation to generate evoked potentials within the tumor, a direct neural connection was found between 4T1 breast tumors and the brain via the vagus nerve [124]. The electrical activity in the tumor was eliminated by chemical sympathectomy [124]. In another study, the presence of sympathetic nerves within the periphery of breast tumor of 4T1 model was described [132].

These findings allow two important conclusions to be drawn. First, activation of vagal fibers can stimulate intra-tumoral sympathetic fibers, leading to an unwanted sympathetic activity within the tumor microenvironment. Second, identifying the factors that drive trans-differentiation of the sensory neuronal phenotype to an adrenergic phenotype (that, in turn, can accelerate cancer growth and metastasis [128,129,133]) may open up new therapeutic opportunities in cancer therapy.

Although the formation of new nerve fibers is called ‘neo-neurogenesis” [120], it might be more accurate to call this process “abnormal neo-neurogenesis”, since these new nerve fibers can promote cancer growth [62]. Exploring the type and neurochemistry of these nerve fibers within the tumor microenvironment in relation to tumor grade and aggressiveness may provide novel patient-tailored treatment options. There are few reports focusing on these issues. For example, progenitors that express doublecortin (a neural migration protein) can initiate autonomic adrenergic neurogenesis in prostate cancer [134]. On the other hand, loss of TP53 drives the reprogramming of tumor-innervating sensory nerves into adrenergic neurons, promoting the growth of head-and-neck carcinomas [121,135]. Intra-tumoral growth of sensory nerves might be induced by brain-derived neurotrophic factor (BDNF). BDNF signals through tropomyosin receptor kinase-B (TRKB) receptors on sensory neurons, and recruits these nerves to establish innervation of the ductal tree during mammary gland development [136].

Although indirectly, the level of TRPV1 expression within tumor tissue may estimate the content of sensory neurons. Expression analysis of tumoral TRPV1 has demonstrated an association between TRPV1 expression and immune response. For instance, in cervical carcinoma, CD8+ T-cells were positively correlated, whereas M2-type macrophages were negatively correlated with TRPV1 expression [137]. Furthermore, decreased TRPV1 expression was associated with increased risk of cervical squamous cell carcinoma [137]. In renal cell carcinoma, TRPV1 expression positively correlated with M1-type anti-tumoral macrophages, and negatively correlated with M2-type tumorigenic macrophages [138]. In agreement, low expression of TRPV1 in renal cell carcinoma is associated with poor clinical outcomes [138].

A recent pan-cancer study further cemented the prognostic value of tumoral TRPV1 expression. For example, TRPV1 expression showed a positive correlation with the ratio of immune-stimulatory over immunosuppressive signatures (CD8+ T cell/PD-L1) in five cancer types [139]. Moreover, TRPV1 expression was found to be markedly lower in late-stage (stage III-IV) cancers, and TRPV1 downregulation correlated with worse overall survival [139]. Although these studies did not directly measure the density of TRPV1-positive sensory neurons in cancer tissues, it can be argued that changes in intra-tumoral TRPV1 predominantly reflects changes in TRPV1-positive nerve fibers within the tumor microenvironment.

In general, a loss of cholinergic/peptidergic nerve fibers is considered a bad prognostic factor. However, this may not be the case for all types of malignancies. For example, human melanoma samples were found to have increased TRPV1-expressing neuronal innervation [140]. When co-cultured with B16F10 murine melanoma cells, TRPV1-positive nociceptors directly extended neurites toward the cancer cells [140]. The tumor-invading sensory neurons express more CGRP and Trka, the receptor for NGF. In melanoma, increased CGRP levels were associated with CD8+ T-cell exhaustion, inhibiting anti-tumoral immunity [140]. These results imply that TRPV1-sensitive sensory innervation may enhance melanoma aggressiveness. If this hypothesis holds true, TRPV1 receptor blockers may have a therapeutic value in melanoma. For example, TRPV1 antagonist creams may be applied to suspicious melanocytic lesions to prevent the development of tumorigenic melanoma.

An issue that needs to be considered is the heterogeneous nature of sensory nerve endings. Melanoma drives sensory nerve fibers from adjacent skin that are polymodal and are involved in pain sensation. Visceral afferents are, however, different in that their principal role is to maintain homeostasis of the internal environment. Pain sensation associated with visceral afferents is often poorly defined and projects to somatic structures, such as skin and muscle (a well-known example is cardiac pain that often projects to the shoulder). Hence, not only the possible alteration in phenotype but also the type of sensory neuron from which cancer-invading nerve fibers sprout might be critical.

## 8. TRPV1: Channel Structure, Epigenetic Regulation and Subcellular Expression

Cryo-electron microscopy and X-ray crystallography have provided important insights into TRPV1 channel structure and function [141,142,143]. In contrast to highly selective cation channels, the selectivity filter of TRPV1 is shallow and dynamic, favoring the influx of larger (e.g., Ca^2+^) or smaller (e.g., Na^+^) cations [143]. This explains the long-recognized “limited selectivity for Ca^2+^” of the TRPV1 channel. Furthermore, TRPV1 has fourfold symmetry with different pore profiles for ligand-bound structures and a vanilloid-binding pocket deep within the membrane bilayer [144]. Resiniferatoxin binding to TRPV1 first opens the intracellular gate, followed by selectivity filter dilation and pore loop rearrangement [144]. This resiniferatoxin-induced conformational “wave” is likely to involve additive structural changes in the channel subdomains [144]. Resiniferatoxin binding to each of these four subunits contributes the same activation energy. The potent small-molecule TRPV1 antagonist, SB-366791, binds to this vanilloid site and acts as an allosteric hTRPV1 inhibitor.

TRPV1 can form functional tetramers both with its splice variants [145] and other TRP channels [146]. TRPV1 splice variants may function as negative modulators of the channel activity. TRPV1/TRPA1 hetero-tetramers are intriguing in that they are shown to possess unique activation profiles: for example, TRPV1 agonists of the home-tetrameric channel may function as antagonists of the hetero-tetrameric channel [147].

Epigenetic regulation of *TRPV1* gene expression is an emerging area of research. In rats, histone H3 acetylation of the *Trpv1* gene promoter region regulated the expression of TRPV1 protein in sensory neurons [148]. In a murine model of diabetic neuropathy, SUMOylation protects the TRPV1 protein from metabolic damage and thus delays the development of neuropathic pain [149]. Phosphorylation by protein kinase C also regulates the activity of the channel protein [150].

The TRPV1 protein is present both in the cell membrane and in subcellular localizations [151]. It was speculated that activated TRPV1 can be recycled into a subcellular organism [152], which may represent a molecular mechanism of desensitization. It is not impossible either that TRPV1 expressed subcellularly (e.g., in mitochondria or the Golgi apparatus) serves distinct biological functions. Indeed, mitochondrial TRPV1 expression was implicated in the proliferation and apoptosis of chronic myeloid leukemia cells [153].

In sensory neurons, TRPV1 most likely functions in the cell membrane to detect heat, protons and other noxious stimuli [154]. In immune cells and cancer, subcellular TRPV1 may be involved in as-yet unsuspected functions. For example, in human breast cancer “non-classical” TRPV1 expression (in the Golgi area) heralds poor prognosis compared to “classical” expression in the cell membrane [155].

## 9. Capsaicin Actions in Immune Cells: On- or Off-Target?

The wide distribution of TRPV1 suggests a diverse function beyond noxious heat sensation and pain perception. Therefore, capsaicin actions in non-neuronal cells were hardly unexpected. However, it is important to keep in mind that capsaicin actions are not necessarily TRPV1-mediated [8]. In fact, these non-TRPV1 mediated capsaicin actions had long clouded the existence of a specific “capsaicin receptor.” For example, capsaicin was reported to stimulate or inhibit the activity of enzymes [156], influence membrane fluidity [157], or interact at voltage-gated ion channels [158], just to cite a few examples of non-specific capsaicin actions.

To provide unequivocal evidence that a capsaicin response is on-target, (1) one should demonstrate the presence of functional TRPV1, and (2) prove that the capsaicin action is absent when TRPV1 is non-functional (e.g., pharmacological blockade or genetic inactivation). For instance, if a capsaicin action is on-target (that is, mediated by TRPV1), it should be absent in TRPV1-null animals. This was not always the case in the carcinogenesis studies. For example, in a mouse skin carcinogenesis experiment with DMBA, capsaicin equally promoted the formation of tumors in TRPV1 wild-type and knock-out animals [45].

Many studies rely on showing the presence of TRPV1 protein by immunostains. Unfortunately, a number of anti-TRPV1 antibodies turned out to be non-specific, producing a similar staining pattern in tissues obtained from wild-type and TRPV1-null animals (Zs. Helyes, personal communication).

Biological effects noted at very high capsaicin concentrations (for example, 100 µM or higher) are especially suspect. In human NK cells, responses to low concentrations of capsaicin (up to 10 µM) were prevented by the TRPV1 antagonists, capsazepine and SB366971 [159]. When the capsaicin concentration exceeded 50 µM, the effect could no longer be antagonized by TRPV1 blockers [159]. These observations imply that capsaicin influences NK cell activity by a mixture of specific and off-target mechanisms, and at high capsaicin concentrations the non-specific actions predominate.

In conclusion, one should take caution when interpreting capsaicin responses noted at high concentrations. However, even by strict criteria, the existence of functional TRPV1 in immune cells was proven beyond doubt. In CD4+ T-cells, capsaicin was shown to stimulate cytokine production: this response was abolished by both pharmacological inhibition and the genetic knock-out of TRPV1 [85].

## 10. Role of TRPV1 Expressing Immune Cells in Cancer

Local and systemic immune responses are critical for inhibiting cancer growth and metastasis [62]. As immune editing states, the ability of the immune surveillance to eliminate cancer cells decreases over time, permitting the growth of highly aggressive tumors. For certain tumors, the TRPV1-mediated neuro-immune regulation of the tumor microenvironment enhances anti-tumoral immunity by inhibiting chronic inflammation [20,62,80,160]. Immune cells express TRPV1, albeit at much lower levels than in sensory neurons [161]. The effect of TRPV1 activation in immune cells on tumor growth and metastasis might be different from that of neuronal TRPV1 (Table 1).

TRPV1 is expressed in mouse and human T cells [85,162], macrophages [163,164], dendritic cells [165,166], Jurkat cells [162], and NK cells [159]. TRPV1 plays an important role in thymocyte differentiation. In accord, the numbers of T-cells are markedly reduced in TRPV1-null mice in both the blood and spleen when compared to wild-type animals [167]. TRPV1 channels are believed to play a critical role in T-cell physiology: for example, rises in TRPV1-mediated intracellular Ca^2+^ concentrations may be required for T-cell activation, proliferation, differentiation, and effector functions [168,169]. In support of this concept, capsaicin increases Ca^2+^ influx and intracellular Ca^2+^ concentration in activated CD4+ T-cells, but not in resting T-cells [170,171]. TRPV1 expressed in CD4+ T-cells is involved in the activation and differentiation of T-cells into Th1 effector cells, inducing an inflammatory response in the murine model of colitis [172]. Furthermore, genetic or pharmacological inhibition of TRPV1 in CD4+ T-cells was also associated with a reduction in airway inflammation in a model of allergic asthma [173]. These observations imply that TRPV1 activation in T-cells may indeed enhance inflammatory responses.

Depending on their duration and concentration, TRPV1 agonists may induce T-cell death. For instance, prolonged exposure to high capsaicin concentrations can induce apoptosis in human peripheral T- and Jurkat cells [171,174]. Similarly, in cultured murine Peyer patch cells in vitro, capsaicin can inhibit the cytokine production by T-cells and reduce T-cell viability [175]. Moreover, TRPV1 levels are significantly increased in T-cells isolated from B16F10 tumor-bearing mice compared to control mice [176]; this is also associated with increased basal intracellular Ca^2+^ levels in T-cells. These findings suggest that TRPV1 agonists may have more profound effects on T-cells in cancer patients, ranging from excessive inflammation to T-cell exhaustion and T-cell death.

The role of TRPV1 in NK cell biology remains unclear. Treatment with a TRPV1 antagonist can increase the population of NK cells, but, at the same time, downregulate the activation of NK/T cells in mice infected with Plasmodium [177]. TRPV1 in human NK cells is functional since 10 µM capsaicin increases intracellular Ca^2+^ concentrations, while pretreatment with the specific TRPV1 antagonist, SB366791, inhibits this effect [159]. (Parenthetically, the further rise in intracellular Ca^2+^ in response to capsaicin concentrations exceeding 50 µM is no longer antagonized by SB366791, indicating a non-specific capsaicin action [159].)

The role of TRPV1 in dendritic cells is not clear either. The available evidence implies an indirect effect of TRPV1 activation on dendritic cells through the neuro-immune axis [118]. In humans, TRPV1 expression significantly increases during in vitro differentiation of monocytes to immature dendritic cells [178]. However, it is not known how TRPV1 expression in NK and dendritic cells may change in cancer patients.

**Table 1 biomolecules-13-00983-t001:** TRPV1 expression in immune cells and its postulated role in carcinogenesis.

Cell Type	TRPV1 Functions	Possible Role in Carcinogenesis
T cells	TRPV1 implicated in TcR-induced Ca^2+^ influx. Capsaicin increases Ca^2+^ influx and intracellular Ca^2+^ concentration in activated CD4+ T-cells, but not in resting T-cells [170,171]. Exposure to prolonged and high concentrations of capsaicin induces apoptosis of human peripheral T- and Jurkat cells [171,174].	TRPV1 levels are significantly increased in T-cells isolated from B16F10 tumor-bearing mice [176]. Possible role in anti-tumoral immune response and T-cell exhaustion requires further studies.
NK cells	Capsaicin increases intracellular Ca^2+^ concentrations in human NK cells while pretreatment with specific TRPV1 antagonists inhibits this effect [159]. TRPV1 antagonist increases the population of NK cells but also downregulates activation of NK/T cells in mice infected with Plasmodium [177].	Not known
Dendritic cells	TRPV1 expression significantly increases during in vitro differentiation from monocytes to immature dendritic cells [178]. Indirect effect of TRPV1 activation on dendritic cells through neuroimmune axis was suggested [118]. Possible role in antigen presentation.	Not known
Mixed leukocyte culture	TRPV1 activation in control mice increases IFN-γ and IL-17, does not alter IL6 release [160].	TRPV1 activation in tumor-bearing mice (breast carcinoma) decreases IFN-γ and increases IL-6 [160].

In the 4T1-related breast carcinoma model, mixed leucocyte cultures of tumor-bearing mice secrete markedly higher levels of IL-6 and TNF-α, and lower levels of IFN-γ [160]. When directly added to in vitro cultured mixed leucocytes, TRPV1 agonists differentially regulate cytokine secretion in tumor-bearing mice compared to controls [160]. Specifically, TRPV1 agonists markedly increase IFN-γ response in control mice, while the opposite effect occurs in tumor-bearing mice [160]. Furthermore, TRPV1 agonists increase IL-17 secretion in control mice, but not in tumor-bearing mice. Lastly, TRPV1 agonists markedly increase IL-6 secretion in tumor-bearing mice, but not in control mice. These observations imply that in the absence of aggressive inflammatory carcinoma, the activation of TRPV1 may enhance immune response against newly formed precancerous lesions by increasing IFN-γ [179] and IL-17 production [180]. On the other hand, the opposite effect may occur in the presence of highly aggressive and inflammatory carcinoma: that is, the activation of TRPV1 in immune cells may create excessive inflammatory conditions [160]. In conclusion, in the presence of inflammatory metastatic carcinoma, doses of TRPV1 agonists high enough to activate TRPV1 in immune cells may have detrimental consequences; this effect may also partly explain the tumor-promoting action of TRPV1 agonists under certain conditions.

## 11. Discussion

An estimated quarter of the world’s population consume capsaicin-flavored food on a daily basis, and many more use capsaicin-containing creams and ointments for topical pain relief. Three large epidemiological studies (one from the US [181], one from Europe [182], and another from China [91]) found a distinct health benefit of eating chili pepper, including lower cancer-related death rates. There are no reports of increased skin cancer risk in users of high-dose capsaicin patches either. Therefore, one may conclude that topical (oral mucosa, GI tract, or skin) exposure to TRPV1 agonists, such as capsaicin, is most likely safe.

A separate question is whether one can block carcinogenesis and/or metastasis formation by manipulating TRPV1-expressing nerves and immune cells. The answer is probably tumor-specific. For example, chronic inflammation is a well-known risk factor of pancreatic carcinoma [183]. Sensory neurons that mediate neurogenic inflammation are believed to play an active role in the initiation and progression of this type of cancer [184]. It was speculated that in the pancreas, chronic inflammation could drive carcinogenesis through both *K-ras* activation and independent of *K-ras* [185]. If so, silencing TRPV1-expressing nerves using capsaicin may protect against pancreatic cancer. This proposal has recently gained experimental support by the protective action of RTX-desensitization in a melanoma model [140]. By contrast, TRPV1-mediated neurogenic inflammation was reported to protect mice from colonic adenocarcinoma [95]. Similarly, activation of TRPV1-positive sensory nerve fibers was shown to enhance anti-tumoral immunity in a breast cancer model [20]. Therefore, capsaicin desensitization may be a “double-edged sword” in the fight against cancer: it may protect against pancreatic cancer and melanoma and, at the same time, conversely increase the risk for colon and breast cancer.

TRPV1-positive sensory nerves might also play a crucial role in the vicious cycle that drives metastatic bone disease [186]. Cancer-associated bone pain is thought to involve sensitization and continuous activation of TRPV1-expressing nerves by the tumor microenvironment [186] (Figure 2). Indeed, in companion dogs with bone cancer, chemical ablation by intrathecal resiniferatoxin of sensory afferents ameliorated the debilitating cancer pain [187,188]. At present, intrathecal resiniferatoxin as a “molecular scalpel” is undergoing clinical trials for permanent analgesia in women with cervical carcinoma metastatic to the bone. If the hypothesis that capsaicin-sensitive afferents drive metastatic bone disease is true, then resiniferatoxin-treated patients are expected to have a reduced tumor burden in addition to pain relief. On the other hand, sensory afferents may play the opposite role in breast carcinoma where capsaicin desensitization was found to promote the spread of metastatic breast cancer [20,73,189]. Taken together, these findings imply that ablation of sensory nerves with resiniferatoxin should not be used for pain relief in women with metastatic mammary carcinoma. Instead, desensitization by perineural TRPV1 agonist administration of nerve fibers might be a better approach for pain relief. In preclinical pain models, perineural resiniferatoxin was shown to achieve lasting conduction analgesia [190].

A third question is the effect (if any) of small-molecule TRPV1 antagonists on carcinogenesis. It is safe to assume that these compounds did not display any carcinogenic potential in animal experiments. In clinical trials, most of these antagonists caused a hyperthermic reaction [190,191,192,193]. Furthermore, minor burn injuries were reported as adverse effects [5]. So far, no concern has been raised with regard to carcinogenesis, but the clinical experience with these compounds is very limited.

TRPV1 activation and/or blockade may have markedly different effects on carcinogenesis depending on both the tumor and its microenvironment. Three large epidemiological studies suggest that TRPV1 activation by culinary capsaicin exposure protects against cancer [91,181,182]. In experimental animals, large doses of a TRPV1 agonist might be tumorigenic [194,195], presumably by activating immune cells. The relevance of these observations in humans is, however, unclear since men are unlikely to be exposed to similarly high TRPV1 agonist doses.

In conclusion, there is good evidence to implicate TRPV1-expressing afferents and immune cells in the bi-directional communication between cancer and its microenvironment. The fine details of this interactions are yet to be elucidated. Results obtained with high capsaicin concentrations have to be carefully reevaluated to exclude the possibility of an off-target action. Further studies are warranted to tailor specific TRPV1-based approaches for cancer therapy as some cancer patients may respond to TRPV1 agonists whereas others may benefit from TRPV1 blockade.

## Figures and Tables

**Figure 1 biomolecules-13-00983-f001:**
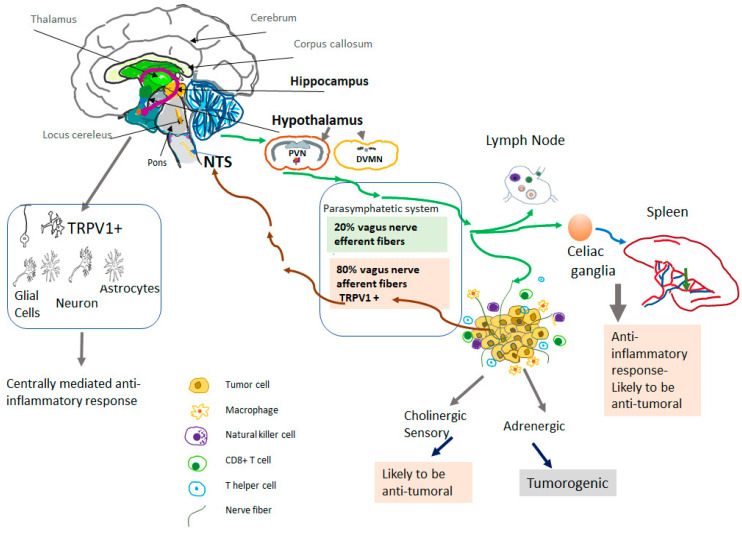
TRPV1-mediated neuro-immune response to cancer. TRPV1 is found primarily in afferent sensory nerve fibers. TRPV1 is also expressed in the central nervous system, mainly in areas that are responsible for controlling the autonomic output to the visceral organs. Specifically, TRPV1 channels were demonstrated in the paraventricular nucleus of the hypothalamus (PVN) and the dorsal motor nucleus of the vagus (DMVN). Besides the hypothalamus, TRPV1 is expressed in the hippocampus and midbrain and the expression is not only found in neurons but also in glial cells and astrocytes. Direct central activity of TRPV1 agonists might be responsible for a systemic anti-inflammatory response. In addition to the activation of vagal TRPV1+ afferent fibers, it may induce a systemic anti-inflammatory response that might be responsible for the anti-tumoral effects of vagal activity. Vagal afferent nerve fibers may activate intra neuronal nerve fibers. NTS-Nucleus Tractus Solitarius.

**Figure 2 biomolecules-13-00983-f002:**
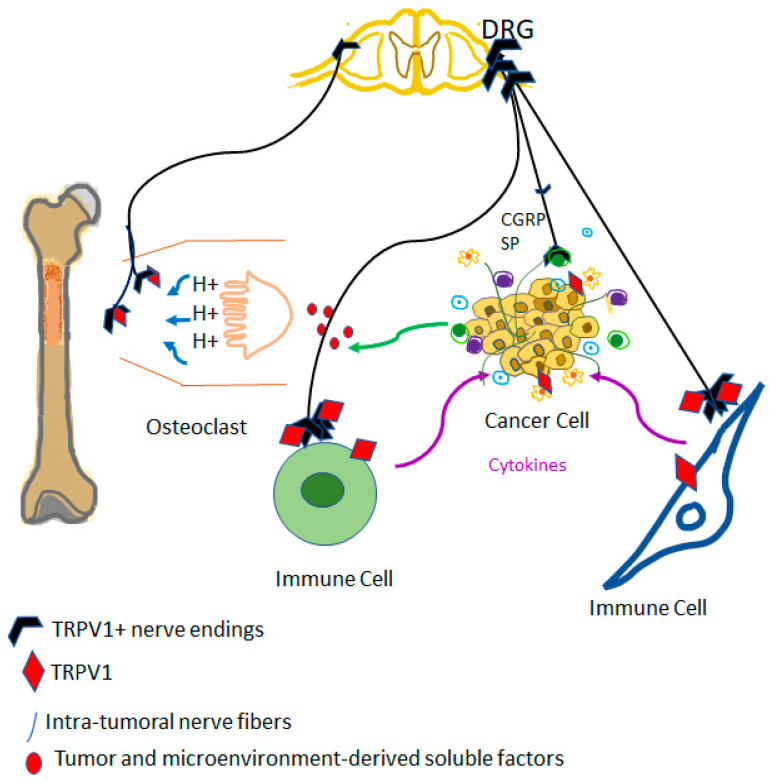
TRPV1 as a driver of the vicious cycle of metastatic bone disease. Cancer cells metastatic to bone may stimulate osteoclasts that, in turn, create an acidic environment. Protons are well-known activators of TRPV1 on sensory afferents; this is perceived as cancer pain in the central nervous system. Furthermore, TRPV1-expressing efferents are sites of release for pro-inflammatory neuropeptides, such as substance P (SP), and calcitonin gene-related peptide (CGRP): these peptides may promote the growth of metastatic cancer by blocking immune cells and increasing blood supply.

## Data Availability

Not applicable.

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
