# Peer review of "Carcinogenesis and Metastasis: Focus on TRPV1-Positive Neurons and Immune Cells"

_biomolecules, 2023, doi:10.3390/biom13060983_

Round 1

Reviewer 2 Report

I appreciate the opportunity for revising this review by Nuray Erin & Arpad Szallasi entitled “Carcinogenesis and metastasis: focus on TRPV1-positive neurons and immune cells”. The paper is well written, didactic and well documented.

However, it is regrettable that the authors did not address the notions of channel structure or specific intracellular expression (plasma membrane versus reticulum).

The biophysical and biochemical characteristics of TRPV1 influence the role of these channels in physiological functions, including immunology.

Perhaps these elements deserve to be included in the discussion or recognized as an editorial boundary chosen by the authors to guide their review.

Round 2

Reviewer 1 Report

The concerns of the reviewer were all addressed in the manuscript as revised.

Reviewer 2 Report

The authors were responsive to the issues I have raised and the manuscript was substantially improved by incorporating the requested corrections.